# The Influence Mechanism of Quantum Well Growth and Annealing Temperature on In Migration and Stress Modulation Behavior

**DOI:** 10.3390/nano14080703

**Published:** 2024-04-18

**Authors:** Luyi Yan, Feng Liang, Jing Yang, Ping Chen, Desheng Jiang, Degang Zhao

**Affiliations:** 1State Key Laboratory of Integrated Optoelectronics, Institute of Semiconductors, Chinese Academy of Sciences, Beijing 100083, China; luyiyan0601@163.com (L.Y.); yangjing333@semi.ac.cn (J.Y.); pchen@semi.ac.cn (P.C.); dsjiang@red.semi.ac.cn (D.J.); 2College of Materials Science and Opto-Electronic Technology, University of Chinese Academy of Sciences, Beijing 100049, China; 3School of Electronic, Electrical and Communication Engineering, University of Chinese Academy of Sciences, Beijing 100049, China

**Keywords:** InGaN/GaN MQWs, migration, temperature characteristics, luminescence performance

## Abstract

This study explores the effects of growth temperature of InGaN/GaN quantum well (QW) layers on indium migration, structural quality, and luminescence properties. It is found that within a specific range, the growth temperature can control the efficiency of In incorporation into QWs and strain energy accumulated in the QW structure, modulating the luminescence efficiency. Temperature-dependent photoluminescence (TDPL) measurements revealed a more pronounced localized state effect in QW samples grown at higher temperatures. Moreover, a too high annealing temperature will enhance indium migration, leading to an increased density of non-radiative recombination centers and a more pronounced quantum-confined Stark effect (QCSE), thereby reducing luminescence intensity. These findings highlight the critical role of thermal management in optimizing the performance of InGaN/GaN MQWs in LEDs and other photoelectronic devices.

## 1. Introduction

Group III–V semiconductor materials and InGaN/GaN multi-quantum wells (MQWs) are widely used in blue and green light-emitting diodes (LEDs) [1,2,3], lasers (LDs) [4,5], solar cells [6], and other optoelectronic devices due to their excellent optoelectronic properties. Due to the complex physical and chemical processes involved in the MOCVD growth of InGaN materials, these processes have a significant impact on the quality of InGaN. Further research is needed on how to achieve high-quality green light emission from InGaN/GaN quantum wells. Green emission requires a high content of In in the used InGaN, which may lead to fluctuations in the In composition and phase separation in quantum well structures [7,8]. In order to avoid the attenuation of luminescence intensity, it is necessary to make the InGaN layer thin during the growth process to reduce defects caused by large strains. Chen et al. [9] found that an appropriate increase in the ammonia flow rate during the growth of quantum wells can effectively enhance the incorporation of In into the quantum wells, and the variations in surface morphology induced by altering the ammonia flow rate result in two distinct luminescence mechanisms in InGaN/GaN quantum wells. Lund et al. [10] achieved the high-quality growth of InGaN films with higher In concentrations by using relaxed N-polar InGaN pseudo-substrates (PSs). However, the influence of temperatures, such as layer growth temperature and annealing treatment temperature on the indium migration mechanism during the growth process of InGaN/GaN multi-quantum wells is still not very clear.

In this study, we investigated the growth of green-light InGaN/GaN MQW samples using the MOCVD method. The migration behavior of the In component in quantum wells under different temperatures was discussed. Through photoluminescence (PL) and electroluminescence (EL) analyses of InGaN/GaN QWs, the effects of well layer growth temperature and annealing temperature on the luminescence characteristics of quantum wells were studied, and quantum well samples with uniform luminescence distribution and localized state distribution were achieved.

## 2. Experiments

A series of green quantum well samples were grown. In the MOCVD growth process of quantum wells, TMIn, TMGa, and ammonia are used as precursors for In, Ga, and N, respectively. Cp_2_Mg and SiH_4_ were used to achieve p-type and n-type doping of GaN layers, and N_2_ and H_2_ were used as carriers during the growth process. N_2_ is used as the carrier gas during the growth of quantum wells, whereas H_2_ is employed for the growth of other structures. This is because H_2_ can degrade the quantum well structure to some extent. At first the GaN buffer layer and n-type GaN layer were grown on the (0001) crystal plane of a sapphire substrate. Then, two cycles of InGaN/GaN quantum wells were grown on the GaN layer at 660 °C, 670 °C, and 680 °C to create 3 samples, named as samples 1, 2, and 3, respectively. The growth temperature of QW layers and quantum barrier (QB) layers was kept consistent. During the MOCVD growth process, thermal annealing was performed on three samples after the growth of the InGaN QW layer and before the growth of the GaN QB layer. The annealing temperature for all three samples was set to 840 °C. Finally, a p-GaN layer and a p-type Ohmic contact layer were grown at the top of the multiple quantum well (MQW)’s structure. Additionally, a sample 4 was grown with an annealing temperature set to 860 °C, while all other growth conditions were kept the same as for sample 3. Among the four samples, samples 1, 2, and 3 and samples 3 and 4 form two different sample series, which are used to investigate the influences of QW growth temperature and annealing temperature on InGaN/GaN MQWs, respectively.

Thomas Swan MOCVD epitaxial equipment (Thomas Swan, Consett, UK) with a tightly coupled nozzle reactor was used in the experimental growth. In the MOCVD system, a synergistic collaboration between multiple temperature regulatory mechanisms is employed to facilitate precise thermal management. This system constitutes a high-precision temperature control framework. Specifically, one component of this arrangement includes a thermocouple temperature sensor in conjunction with a temperature controller (EUROTHERM, Baulkham Hills, NSW, Australia), which are collectively responsible for the measurement and modulation of the heater’s temperature. Additionally, the MOCVD system is equipped with a commercial temperature monitoring system from LayTEC (Berlin, Germany), which operates on the principle of an optical pyrometer system. The integration of these components ensures the maintenance of minimal temperature fluctuations throughout the operation. The structural parameter information of samples is obtained through high-resolution X-ray diffraction (HRXRD) with a Cu K_α_ source of radiation and is conducted using a Rigaku SmartLab X-ray diffractometer (Rigaku, Tokyo, Japan) at (0002) reflection. The lattice relaxation of the samples is determined through reciprocal space mapping (RSM). Temperature-dependent PL (TDPL) and EL testing is used to study the luminescence characteristics of quantum wells. PL testing uses a 405 nm laser and a grating spectrometer. EL testing is conducted under direct current injection using an Ocean Optics HR 2000 high-resolution spectrometer (Ocean Optics, Dunedin, FL, USA). Micro-photoluminescence images of samples are examined, which are excited under a 405 nm laser and are acquired using Nikon A1 confocal optical system (Nikon, Tokyo, Japan).

## 3. Results and Discussion

Figure 1 shows the PL emission spectra of samples 1, 2, and 3 at room temperature. It can be seen that the luminescence intensity of sample 3 is much higher than those of samples 1 and 2. The peak PL wavelengths of the three samples are 548.7 nm, 529.9 nm, and 525.4 nm, respectively. As the growth temperature of the well layers increases, the luminescence peak of the quantum well undergoes a significant blue shift. During the growth process of quantum wells, all conditions remain constant except the growth temperature, so the change in luminescence peak wavelength comes from the change in quantum well structure induced by the growth temperature. Furthermore, the analysis of quantum well structures was conducted by HRXRD. The 2θ/ω scanning results of three samples are shown in Figure 2, where the main GaN peaks of the three curves are all at 34.5°, while the -1st and -2nd satellite peaks shift to a certain extent, showing that a change in growth temperature indeed causes changes in the structural parameter of the quantum well. The XRD scanning curves are fitted using GlobalFit Version 2.0.6.0 to obtain the sample structural parameters in Table 1. With the increase in the growth temperature of the well layer, there is a slight decrease in the thickness of the well layer and a reduction in the In concentration within the quantum well.

The increase in growth temperature would lead to an increase in the rate of In incorporation into the quantum well layer, and the In component in the quantum well layer should be increased according to literature [11]. However, the results obtained are exactly the opposite. In fact, previous studies have indicated that the growth of InGaN layers is primarily governed by a combination of the mass transport mechanism and In desorption mechanism, with In desorption playing a dominant role in the growth at higher temperatures [12]. In our experiment, the duration of growth for the InGaN layers of all three samples was the same. The thickness of sample 1 suggests that the actual growth rate for sample 1 was only slightly higher than those of other two samples. At elevated growth temperatures, the rate of In desorption increases with temperature, leading to a decrease in the indium concentration within the quantum well layers. Correspondingly, a reduced availability of In participating in the growth results in a slowdown of the actual growth rate of the InGaN layer.

Figure 3 shows confocal fluorescence microscopy images of samples 1, 2, and 3 under the same laser beam excitation. It can be seen that there are a large number of uneven maze-shaped patterns in the field of view of sample 1, accompanied by a large number of black spots. However, this phenomenon was significantly alleviated in the two samples with higher growth temperatures. According to previous studies [8], the black spots observed in confocal fluorescence microscopy are due to the separation of In atoms from the interior of the quantum well layer during the growth and formation of In clusters on the surface of the quantum wells. Due to the significant lattice mismatch between InGaN and GaN, high strain energy is stored in the quantum wells, which promotes the aggregation of In atoms towards the surface in InGaN/GaN quantum wells, forming In clusters to reduce strain energy. In the aforementioned XRD analysis, it was found that as the growth temperature increases, the content of the In component in the quantum well layer decreases, which reduces strain energy in the quantum wells and suppresses the In component’s pulling effect, thus suppressing the formation of In clusters. At the same time, a higher growth temperature also increases the mobility of In on the surface of the quantum wells, resulting in less In segregation, and the observed light field becomes more uniform.

Three samples were subjected to variable-temperature PL testing under 405 nm laser excitation. Considering the possible influence of the disturbance effect caused by Fabry–Perot interference on the experimental data, Gaussian function fitting was used to preprocess the PL data. Figure 4 shows the temperature-dependent peak positions for samples grown at different quantum well temperatures. All three curves show a trend of red shift followed by blue shift under temperature increase, indicating that the main luminescence mechanism of the three MQWs samples is localized state luminescence. At low temperatures, the emission peak position shifts towards longer wavelengths due to the migration of charge carriers from shallow localized states to deep localized states within the quantum wells. As the temperature continues to rise, all carriers in the shallow localized state enter the deeper localized state. At this point, as the temperature continues to increase, more carriers gain energy and transit to energy levels closer to the band edge, resulting in a blue shift [13]. The peak position in sample 2 (Figure 3b) has undergone a red shift again, which is mainly due to the narrowing effect of the band gap caused by the increase in temperature. According to Eliseev’s tail state model, the relationship between the emission peak position and temperature during variable temperature PL testing can be summarized as follows when the ambient temperature is above approximately 125 K [14]:$$E_{0}(T)=E_{0}(0)-\frac{\alpha T^{2}}{T+\beta }-\frac{\sigma^{2}}{k_{B}T}$$
where *E*_0_(*T*) represents the luminescence peak position of the material at temperature *T*, *E*_0_(0) signifies the energy level of the localized state center, *α* and *β* are fitting parameters, *k_B_* is the Boltzmann constant, and *σ* denotes the distribution width of the localized states. The fitting curves are also depicted in Figure 4 by the pink lines. According to the fitting results, the *σ* values for the quantum well samples 1, 2, and 3 grown at 660 °C, 670 °C, and 680 °C are 0.11 meV, 6.25 meV, and 16.17 meV, respectively. This indicates that sample 3 exhibits the most pronounced localized state effect, which hinders the transfer of carriers between localized states. This conclusion is shown again in the relationship between the full width at half maximum (FWHM) of the emission peak and temperature: the deep localized state of sample 3 effectively inhibits the migration of carriers to other localized states, making the emission more centralized at the main peak, which is reflected by the FWHM of the emission peak being much narrower than those of the other two samples, as shown in Figure 5.

In order to explore the effect of annealing temperature, sample 4, which was annealed at a higher temperature of 860 °C, was introduced and compared with sample 3, which was annealed at 840 °C. Figure 6 shows the XRD 2θ/ω scanning curves of two samples, and they are fitted with the Globalfit program to obtain the content of the In component and the thickness of the QW and QB layers of the two samples, as shown in Table 2. It can be seen that both samples have a significant GaN main peak, as well as −1, −2, and −3 satellite peaks caused by quantum wells. The −3 satellite peak of sample 4 is more blurred on the low angle side, which may be caused by deterioration in the interfaces of the quantum wells. It is found that with the increase of annealing temperature, the periodic thickness of the quantum well decreases, and the content of In component increases slightly. Figure 7 presents the reciprocal space mapping (RSM) images of samples 3 and 4. Upon comparison, it is evident that both samples maintained coherent growth throughout the process, with essentially no lattice relaxation. Due to the significant difference in lattice constants between GaN and InGaN, the coherently grown InGaN/GaN quantum well structures will accumulate coherent strain energy, which can influence the subsequent growth of quantum well structure, manifesting macroscopically as a reduction in layer thickness. The varying degrees of thinning observed in the QW and QB layers of sample 4 could be due to this reason. A too high annealing temperature exacerbates the instability of quantum wells with accumulated strain energy, making the influence of the strain energy on the subsequent growth of quantum wells even more pronounced. When the InGaN layer is exposed to a higher temperature, In segregation is enhanced, leading to a decrease in the In content. Concurrently, an appropriate increase in the annealing temperature facilitates the redistribution of In within the quantum well layers. This may promote the entrance of In atoms into the quantum wells, resulting in the slightly higher In concentration in sample 4. The analysis of RSM images reveals that within the temperature range studied in the experiments, variations in annealing temperature do not induce significant lattice relaxation. However, they do exert a certain influence on the subsequent growth rate of quantum wells.

Furthermore, EL tests were conducted on samples 3 and 4 which had undergone different annealing heat treatments. The EL spectra of sample 3 and sample 4 (not provided) were recorded and compared at an injection current of 10 mA. It was found that when the injection current increased, the emission peak of sample 4 had only a small shift to s shorter wavelength compared to sample 3, and its emission intensity was slightly lower than that of sample 3. This is due to the combined effect of a thinner quantum well thickness and a higher In concentration. The spectral lines of the EL emission peak under different injection currents are fitted with the Gaussian function to obtain the peak energy and FWHM. Their dependencies on the injection current are shown in Figure 8a,b. At low current injection levels, with the increase in the injection current, the luminescence peak position of the quantum well has a significant blue shift, which is caused by the increasing Coulomb shielding effect of the carriers. The applied electric field compensates the polarizing electric field in the quantum wells, which reduces the tilt of the energy band, resulting in a blue shift. Therefore, the relative value of the electric polarization field can be determined according to the blue shift. The blue shifts of samples 3 and 4 are 46.4 meV and 64.18 meV, respectively. This means that the polarization effect in the quantum well region of sample 4 is stronger. The enhanced polarization effect leads to a stronger quantum confinement Stark effect (QCSE) [15]. On one hand, the emission peak position is red-shifted with the increasing injection current. On the other hand, the spatial coincidence of electron and hole wave functions is reduced due to the energy band tilt, and the radiative recombination luminescence will be weakened. The change trend of the FWHM of the two samples is basically the same. The decrease in FWHM at low current injection level is due to the suppression of the QCSE by the Coulomb shielding effect. With the further increase in the current, the emission peak broadens, which may be due to the effect of the local states caused by the uneven distribution of In in quantum well layers.

Figure 9 shows the PL test results of the two samples at room temperature. It is obvious that the luminous intensity of sample 4 is only about two-thirds of that of sample 3. On one hand, for the quantum well samples treated at higher annealing temperature, more remarkable strain relaxation occurs during annealing, and more defect states are introduced into the quantum well. These defect states will act as non-radiative recombination centers to capture carriers and suppress the luminescence efficiency of radiative recombination. On the other hand, the bond energy of the In-N bond is very low. The higher-temperature annealing may also lead to more damages in quantum wells and affect their luminescence properties. In order to further obtain the information of the non-radiative recombination centers in the samples, the luminous integral PL intensities of the two QW samples were quantitatively calculated after removing the background, and their relationship with reciprocal temperature was derived as shown in Figure 10. The data were fitted with Arrhenius function as follows:$$I(T)=\frac{1}{1+\sum C_{i}\exp(-E_{i}/k_{B}T)}$$
where *I*(*T*) represents the normalized integrated PL intensity of the sample at a given temperature *T*, *C_i_* is a fitting parameter related to the density of non-radiative recombination centers, and *E_i_* is the activation energy of the non-radiative recombination center [16,17]. We found that when there are two types of non-radiative recombination centers, the fitting curve matches the experimental values best, which satisfies
$$I(T)=\frac{1}{1+C_{1}\exp(-\frac{E_{1}}{k_{B}T})+C_{2}\exp(-\frac{E_{2}}{k_{B}T})}$$

The fitting curve is also shown in Figure 10. The values of the fitting parameters obtained are given in Table 3. It is shown that the activation energies of the non-radiative recombination centers with lower activation energies in the two samples are very similar. They are 22.79 meV and 20.12 meV, respectively. The values of fitting parameter *C*_1_ of the two samples are also very close, while the fitting parameter *C*_2_ of sample 3 is much lower than that of sample 4. Because the values of *C*_2_ related to the density of the non-radiative recombination centers of the two samples are much higher than *C*_1_, *C*_2_ is a more important factor affecting the luminescence characteristics of the quantum wells; the non-radiative recombination activation energies of *C*_2_ are 129.52 meV and 104.83 meV for sample 3 and sample 4, respectively. When the annealing temperature increases from 840 °C to 860 °C, the density of non-radiative recombination centers increases significantly. These non-radiative recombination centers come from the degradation of quantum wells during the annealing process as mentioned above. At the same time, the activation energy of these kinds of recombination centers in sample 4 is also relatively lower, which means they are activated at a lower temperature to capture carriers and suppress the luminescence of the quantum wells. As can also be observed from the confocal fluorescence microscope images of samples 3 and 4 in Figure 11, sample 4 exhibits more pronounced dark spot defects. It is worth mentioning that both samples were excited under the same laser power at 405 nm, and all parameters of the confocal fluorescence microscope were kept constant. However, the photoluminescent brightness of sample 4 was significantly weaker than sample 3, indicating that carriers in sample 4 are more prone to being captured by non-radiative recombination centers, leading to a reduction in the sample’s luminescent intensity.

## 4. Conclusions

By changing the growth and annealing temperatures of InGaN/GaN quantum well layers, the migration behavior of In in quantum wells and its influence on the structural quality and luminescence properties of quantum wells were studied. Within the studied growth temperature range in our research, the growth temperature of the quantum wells modulates the In desorption process from the quantum wells, thereby suppressing the net incorporation efficiency of indium. The reduction in the strain energy stored in the quantum well structures leads to an enhancement in the luminescence efficiency of the quantum wells. It was found by TDPL that under high-growth-temperature conditions, the localized state effect is more significant. Higher annealing temperatures exacerbate the limiting effect of strain energy on the QW growth rate and intensify the migration behavior of In into QW. At the same time, a too high annealing temperature will increase the density of non-radiative recombination centers within the QW, leading to an increased emission wavelength and a more pronounced QCSE, which weakens the luminescent performance of the quantum wells.

## Figures and Tables

**Figure 1 nanomaterials-14-00703-f001:**
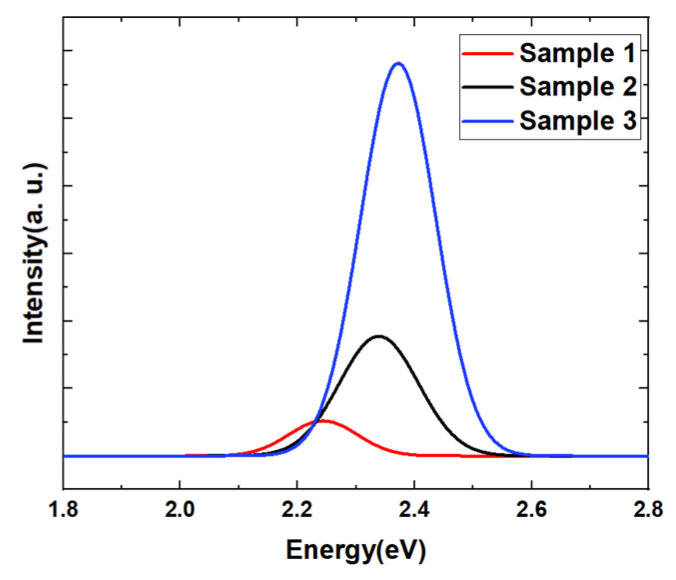
Room temperature PL test results of samples 1, 2, and 3 in which well layers are grown at different temperatures.

**Figure 2 nanomaterials-14-00703-f002:**
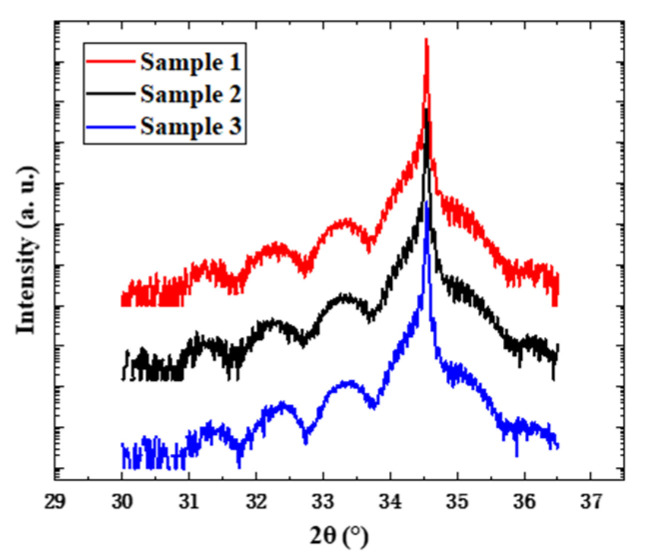
XRD 2θ/ω scanning curves of samples 1, 2, and 3 grown at different quantum well growth temperatures.

**Figure 3 nanomaterials-14-00703-f003:**

The confocal fluorescence microscopy images of the three samples under the same laser gain, (**a**) Sample 1, (**b**) Sample 2, (**c**) Sample 3.

**Figure 4 nanomaterials-14-00703-f004:**
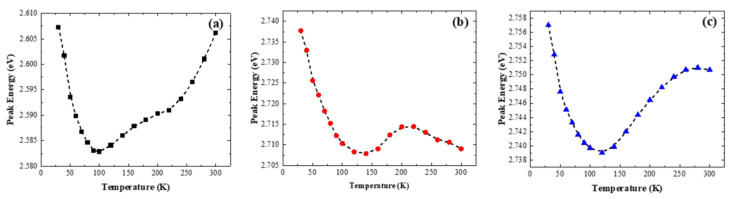
Temperature dependence curve of PL peak position of three samples, 1 (**a**), 2 (**b**), and 3 (**c**).

**Figure 5 nanomaterials-14-00703-f005:**
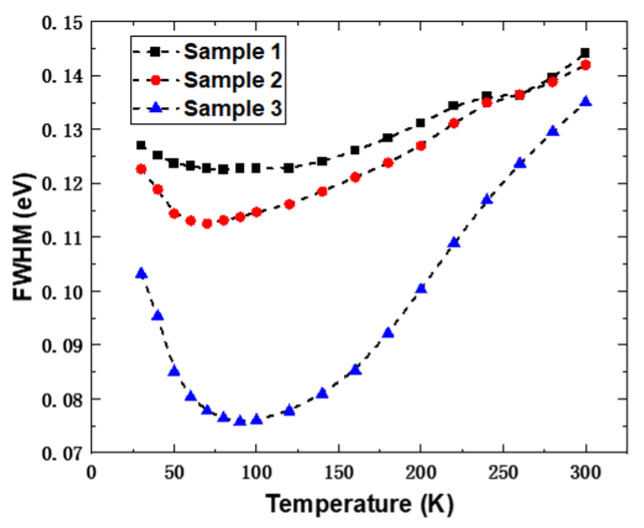
PL peak FWHM as a function of temperature for samples 1 (black), 2 (red), and 3 (blue).

**Figure 6 nanomaterials-14-00703-f006:**
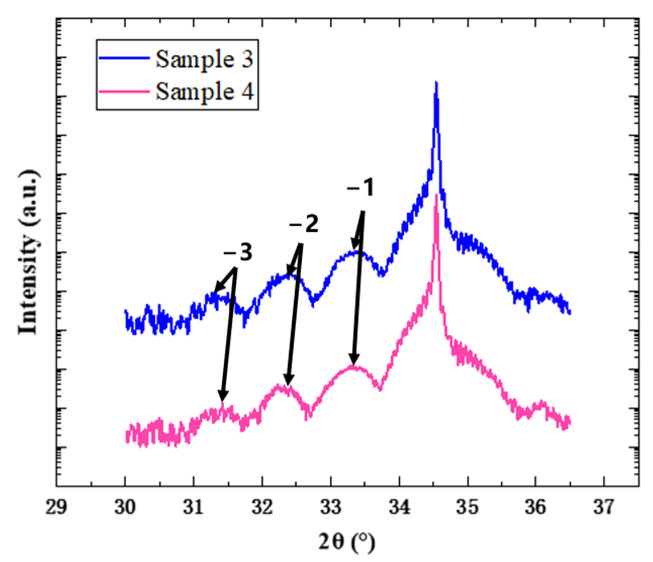
XRD 2θ/ω scanning curves of samples 3 (blue) and 4 (pink) after treatment at different annealing temperatures.

**Figure 7 nanomaterials-14-00703-f007:**
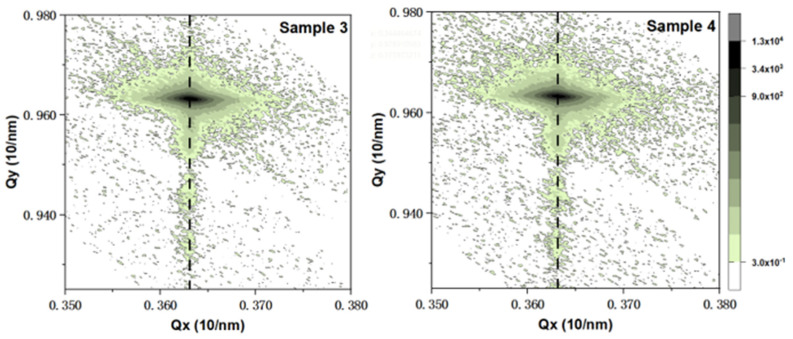
Reciprocal space mapping (RSM) images of (1 0 −1 5) planes of samples 3 and 4.

**Figure 8 nanomaterials-14-00703-f008:**
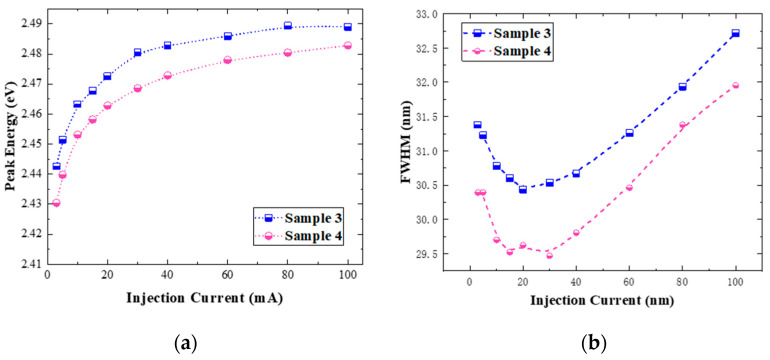
(**a**) Peak energy variation with injection current; (**b**) FWHM variation with injection current for samples 3 (blue) and 4 (pink).

**Figure 9 nanomaterials-14-00703-f009:**
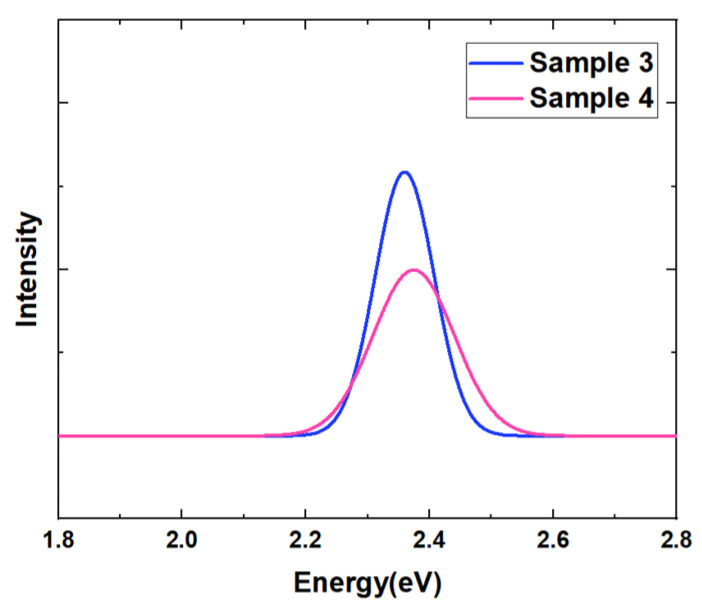
Room temperature PL spectra of multiple quantum wells: sample 3 (blue) and sample 4 (pink).

**Figure 10 nanomaterials-14-00703-f010:**
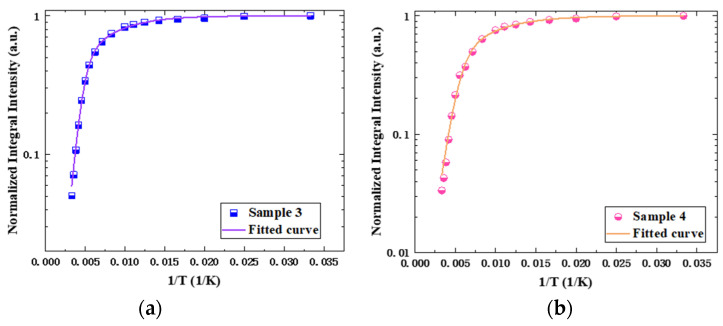
Fitting curve of normalized integral intensity with temperature variation: (**a**) sample 3 and (**b**) sample 4.

**Figure 11 nanomaterials-14-00703-f011:**
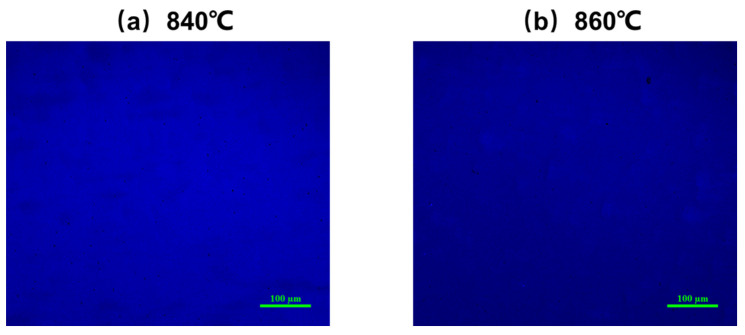
The confocal fluorescence microscopy images of sample 3 (**a**) and sample 4 (**b**).

**Table 1 nanomaterials-14-00703-t001:** Structural parameters of green light quantum well samples.

Sample Number	QW Growth Temperature/°C	Annealing Temperature/°C	QW Layer Thickness/nm	In Content
Sample 1	660	840	2.61	16.0%
Sample 2	670		2.48	15.3%
Sample 3	680		2.55	14.4%

**Table 2 nanomaterials-14-00703-t002:** Structural parameters of green-light quantum well samples.

Sample Number	QW Growth Temperature/°C	QW Growth Temperature/°C	QW Layer Thickness/nm	QB Layer Thickness/nm	In Content
Sample 3	680	840	2.55	6.85	14.4%
Sample 4		860	2.47	6.47	14.9%

**Table 3 nanomaterials-14-00703-t003:** Fitting results of *C_i_* and *E_i_*.

Sample	*C* _1_	*E*_1_/meV	*C* _2_	*E*_2_/meV
Sample 3	3.18	22.79	814	129.52
Sample 4	3.47	20.12	2215	104.83

## Data Availability

The data that support the findings of this study are available from the corresponding author upon reasonable request.

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
