# Peer review of "The Influence Mechanism of Quantum Well Growth and Annealing Temperature on In Migration and Stress Modulation Behavior"

_nanomaterials, 2024, doi:10.3390/nano14080703_

Round 1

Reviewer 1 Report (Previous Reviewer 1)

Comments and Suggestions for Authors

Thank you for taking my comments into account.

However, I would like to ask you to improve the quality of the RLM (Fig. 7) - maybe in color and why does the data go out of the frame? 

and correct the caption of fig. 7 -  (105) - it should be 4 indexes (10-15) as in the entire manuscript.

Author Response

Thank you for your letter and for the reviewer’s comments on our manuscript titled “The Influence Mechanism of Quantum Well Growth and Annealing Temperature on In Migration and Stress modulation Behavior” (ID: nanomaterials-2960774). The following are responses to the questions you raised:

  1. Comment:  I would like to ask you to improve the quality of the RLM (Fig. 7) - maybe in color and why does the data go out of the frame? 

Response: In addressing the issue of errors observed in the RSM images, we have identified that the RSM images did not successfully convert to the appropriate format during the process of translating to the PDF version. In the resubmitted manuscript, we have revised the formatting, and the preview image provided below is from the PDF version of our manuscript, which clearly demonstrates the color distribution within the images, ensuring that no data points are displaced from the axes.

  1. Comment: Correct the caption of fig. 7 -  (105) - it should be 4 indexes (10-15) as in the entire manuscript.

Response: The errors pertaining to the crystal plane indices have also been corrected in the manuscript, as reflected in the aforementioned figure.

Reviewer 2 Report (New Reviewer)

Comments and Suggestions for Authors

Thank you for incorporating all the suggestions and for the response to referee comments. I think the paper is ready for publication after a few minor changes: All the figures are very poor quality and it is very hard to read the axes labels for many of them. For example Fig. 5 the x and y axis are not readable. Perhaps this is because of how the figures were exported to pdf in the journal format? I would request that all figures be made legible before the paper is published.

The additional details about process/experiment are now clearer but further details on exactly how temperature was measured should be added to the experimental section of the paper. If a pyrometer was used, then what was the wavelength of the pyrometer and how was the pyrometer calibrated? Since the paper relies on very small temperature changes (10C) and annealing, it is important to explain to readers how temperature was measured so they can reproduce the study.

Line 254: Equation for arrhenius function seems to have an artifact. Please reconfirm that it is correct.

Author Response

Thank you for your letter and for the reviewer’s comments on our manuscript titled “The Influence Mechanism of Quantum Well Growth and Annealing Temperature on In Migration and Stress modulation Behavior” (ID: nanomaterials-2960774). The following are responses to the questions you raised:

  1. Comment: All the figures are very poor quality and it is very hard to read the axes labels for many of them. For example, Fig. 5 the x and y axis are not readable. Perhaps this is because of how the figures were exported to pdf in the journal format? I would request that all figures be made legible before the paper is published.

Response: Regarding the issue of images clarity, the repeated format conversions had led to a significant degradation in the sharpness of the images. We have updated all data images, and you will find improved clarity in the versions presented within our resubmitted manuscript.

  1. Comment: The additional details about process/experiment are now clearer but further details on exactly how temperature was measured should be added to the experimental section of the paper. If a pyrometer was used, then what was the wavelength of the pyrometer and how was the pyrometer calibrated? Since the paper relies on very small temperature changes (10C) and annealing, it is important to explain to readers how temperature was measured so they can reproduce the study.

Response: In the revised manuscript, how temperature was measured has be added to the experimental section. In MOCVD system, multiple temperature control systems work together to ensure the temperature changes, which is a high precision temperature control system. On one side, A thermocouple temperature detector and a temperature controller (EUROTHERM) are equipped to measure temperature and control heater. Moreover, a commercial temperature monitoring system of LayTEC is also equipped in MOCVD, which is optical pyrometer system. Thus, a small temperature change is guaranteed in this work. The corresponding content has been supplemented in the manuscript, which you can view in lines 74-83 of the manuscript.

  1. Comment: Line 254: Equation for arrhenius function seems to have an artifact. Please reconfirm that it is correct.

Response: We have taken note of the artifact that appeared in the PDF version of the Arrhenius function. In the version we are resubmitting, this has been corrected.

Reviewer 3 Report (New Reviewer)

Comments and Suggestions for Authors

Referee report on manuscript “The Influence Mechanism of Quantum Well Growth and Annealing Temperature on In Migration and Stress modulation Behavior

The article probably contains some new results that may be recommended for publication, but only after its major improvement.

1. I would like to understand why this article was submitted to this journal if the word nano does not appear in the text of the article at all.

2. 17 references, most of which are old, this is not enough to assess the novelty and relevance of the work.

3,  Fig.1.  It would be useful to analyze the given spectra in detail, decomposing them into Gaussian components. See for details (Brik, M. G., Srivastava, A. M., & Popov, A. I. (2022). A few common misconceptions in the interpretation of experimental spectroscopic data. Optical Materials127, 112276.)

4. almost all drawings require improvement because many details are faded and indistinguishable.

5. Table 1. The last column needs error bars and corresponding information in the text.

6. Table 2. The same requirements are also for the last three columns.

7. Figure 9. Again, these spectra need to be deconvoluted into Gaussian components.

8. Finally,  for correctness, the temperature dependence must be analyzed for each Gaussian component of complex luminescence spectra.

Author Response

Dear Reviewer,

Thank you for your letter and for the reviewer’s comments on our manuscript titled “The Influence Mechanism of Quantum Well Growth and Annealing Temperature on In Migration and Stress modulation Behavior” (ID: nanomaterials-2960774). The following are responses to the questions you raised:

  1. Comment:I would like to understand why this article was submitted to this journal if the word nano does not appear in the text of the article at all.

Response: In fact, within our sample structure, the thickness of both the GaN and InGaN layers that form the quantum well are on the nanoscale. The issues we investigated are based on the structure of the quantum well at the nanoscale. Moreover, during the growth process of the quantum well, the growth rate is also maintained at the nanoscale. In summary, we think that our manuscript is suitable for publication in nanomaterials. There are several articales published on nanomaterials with similar research directions that can serve as references for your consideration.

  1. Comment: 17 references, most of which are old, this is not enough to assess the novelty and relevance of the work.

Response: Thank you for your feedback regarding the references. The early literature cited in our manuscript serves as foundational or seminal work that provides important theoretical underpinnings for our analysis. In response to your comments, we have also incorporated several recent references to support the timeliness of our research direction, which you will find updated in the reference section.

  1. Comment:1.  It would be useful to analyze the given spectra in detail, decomposing them into Gaussian components. See for details (Brik, M. G., Srivastava, A. M., & Popov, A. I. (2022). A few common misconceptions in the interpretation of experimental spectroscopic data. Optical Materials127, 112276.) & Figure 9. Again, these spectra need to be deconvoluted into Gaussian components.

Response: Your suggestion for revision is valid. After reviewing the reference you provided, I recognize that there were indeed issue with the way the room-temperature PL curves were treated in our manuscript. In the resubmitted manuscript, these have been corrected by decomposing them into Gaussian components, as you can inspect in the accompanying thumbnail images or within the main text of manuscript.

  1. Comment: Almost all drawings require improvement because many details are faded and indistinguishable.

Response: Regarding the issue of images clarity, the repeated format conversions had led to a significant degradation in the sharpness of the images. We have updated all data images, and you will find improved clarity in the versions presented within our resubmitted manuscript.

  1. Comment: Table 1. The last column needs error bars and corresponding information in the text. & Table 2. The same requirements are also for the last three columns.

Response: In our manuscript, the fitting information presented in the two tables does not include the error values of the fits. This likely stems from the software GlobalFit 's emphasis on optimizing fitting parameters, which in our analysis refer to the epilayer thickness and In composition, rather than on error analysis, hence the software does not directly provide error value.

  1. Comment: Finally,  for correctness, the temperature dependence must be analyzed for each Gaussian component of complex luminescence spectra.

Response: In our manuscript, when analyzing the effects of growth temperature on the luminescence properties and In migration behavior of quantum wells, we conducted a detailed analysis of the Gaussian components for the first three samples. However, while studying the impact of annealing temperature, we observed that variations in strain energy and defects were more pronounced. At this juncture, analyzing the Gaussian components of the PL spectra did not seem to yield additional relevant conclusions. Consequently, this part of the analysis was not presented in the manuscript.

Thank you once again for your valuable comments and suggested revisions to our manuscript.

Round 2

Reviewer 3 Report (New Reviewer)

Comments and Suggestions for Authors

After successful revision this manuscript can be accepted 

This manuscript is a resubmission of an earlier submission. The following is a list of the peer review reports and author responses from that submission.

Round 1

Reviewer 1 Report

Comments and Suggestions for Authors

Dear authors,

To correctly analyse the XRD results (XRD curves) in this type of lattice mismatched layers, a parameter related to layer relaxation is needed. The presented results show nowhere the degree to which the layer is relaxed/strained. This can be determined on the basis of reciprocal lattice maps of asymmetric planes, e.g. (01.5) (11.4). The degree of relaxation also affects the composition of the InGaN layer. Maybe this is the reason why your growth behaves differently compared to the literature - see 99-101.

Therefore, please provide the reciprocal lattice maps and, based on them, the relaxation values of the obtained layers.

Moreover, the description of the equipment and experiment lacks details; do you perform an omega/2theta scan with an analyser or an open detector?

If it is an omega/2theta scan, the X-axis should be Omega not 2Theta, unless it is a 2theta/omega scan.

Generally, the text lacks a description of the experiments; please complete it.

Reviewer 2 Report

Comments and Suggestions for Authors

This study explores the effects of growth temperature of InGaN/GaN quantum well bilayers, as well as the effect of annealing temperature on the indium migration, structural quality, and luminescence properties. Increasing the annealing temperature induces strain relaxation and introduces more defect states into the quantum well structure, which become new non-radiative recombination centers which suppress the radiative recombination and reduce luminescence intensity of quantum wells.

The paper requires a major revision and should be completed with some evidence for non-radiative recombination centers before publication for the following reasons:

1.         Abbreviations: should be defined at first occurrence. Some of them are not defined at all (e.g. TDPL…).

2.         Experimental. Equipment should be defined: both deposition, annealing equipment. Also comment the way of temperature control: how was it possible to control and measure it better than 10 degrees accuracy? Mention confocal microscopy and its basic parameters! What was the deposition temperature of sample 4 (only the annealing temperature is defined)?

3.         The quantum well is a bilayer. How can you characterize it with one structure parameter (Table 1) with thickness dimension? What is the "well width" parameter? Is it the thickness of InGaN layer? How was it possible to determine the In content from the XRD measurement?

4          The authors argue that high annealing temperature induces indium migration and strain relaxation introducing more defect states into the quantum well structure, which become new non-radiative recombination centers. These statements would require structural evidence (e.g. EDS maps) to prove the In migration and reveal the nature of the formed defects. They should provide information about the nature of the non-radiative recombination centers. These must be quite well defined features if they have so well defined activation energy (22.79 meV and 20.12 meV). Without structural evidence and description of the non-radiative recombination centers the argumentation is very speculative.

5.         The English of the paper is “Chinese” sometimes. Is annoying to read “well layers” (“quantum well layers” or “QW layers” would be better). Similarly, “red shift” and “blue shift” are also strange. One could write instead “the peak shifted to longer/shorter wavelengths”. Some most significant mistakes and suggestions are indicated in the attached pdf. The marks in the pdf are not a complete list of their mistakes. A professional English language editing would be beneficial.

Comments on the Quality of English Language

The English of the paper is “Chinese” sometimes. Is annoying to read “well layers” (“quantum well layers” or “QW layers” would be better). Similarly, “red shift” and “blue shift” are also strange. One could write instead “the peak shifted to longer/shorter wavelengths”. Some most significant mistakes and suggestions are indicated in the attached pdf. The marks in the pdf are not a complete list of their mistakes. A professional English language editing would be beneficial.